# The Re-Emergence of Hepatitis E Virus in Europe and Vaccine Development

**DOI:** 10.3390/v15071558

**Published:** 2023-07-16

**Authors:** Gergana Zahmanova, Katerina Takova, Valeria Tonova, Tsvetoslav Koynarski, Laura L. Lukov, Ivan Minkov, Maria Pishmisheva, Stanislav Kotsev, Ilia Tsachev, Magdalena Baymakova, Anton P. Andonov

**Affiliations:** 1Department of Plant Physiology and Molecular Biology, University of Plovdiv, 4000 Plovdiv, Bulgaria; katerina.takova@uni-plovdiv.bg (K.T.);; 2Department of Technology Transfer and IP Management, Center of Plant Systems Biology and Biotechnology, 4000 Plovdiv, Bulgaria; 3Department of Animal Genetics, Faculty of Veterinary Medicine, Trakia University, 6000 Stara Zagora, Bulgaria; 4Faculty of Sciences, Brigham Young University—Hawaii, Laie, HI 96762, USA; llukov@go.byuh.edu; 5Institute of Molecular Biology and Biotechnologies, 4108 Markovo, Bulgaria; 6Department of Infectious Diseases, Pazardzhik Multiprofile Hospital for Active Treatment, 4400 Pazardzhik, Bulgaria; 7Department of Microbiology, Infectious and Parasitic Diseases, Faculty of Veterinary Medicine, Trakia University, 6000 Stara Zagora, Bulgaria; ilia_tsachev@abv.bg; 8Department of Infectious Diseases, Military Medical Academy, 1606 Sofia, Bulgaria; 9Department of Medical Microbiology and Infectious Diseases, Max Rady College of Medicine, University of Manitoba, Winnipeg, MB R3T 2N2, Canada

**Keywords:** hepatitis E virus, emerging disease, HEV transmission, HEV epidemiological studies, HEV vaccines, HEV control and prevention, zoonosis

## Abstract

Hepatitis E virus (HEV) is one of the leading causes of acute viral hepatitis. Transmission of HEV mainly occurs via the fecal-oral route (ingesting contaminated water or food) or by contact with infected animals and their raw meat products. Some animals, such as pigs, wild boars, sheep, goats, rabbits, camels, rats, etc., are natural reservoirs of HEV, which places people in close contact with them at increased risk of HEV disease. Although hepatitis E is a self-limiting infection, it could also lead to severe illness, particularly among pregnant women, or chronic infection in immunocompromised people. A growing number of studies point out that HEV can be classified as a re-emerging virus in developed countries. Preventative efforts are needed to reduce the incidence of acute and chronic hepatitis E in non-endemic and endemic countries. There is a recombinant HEV vaccine, but it is approved for use and commercially available only in China and Pakistan. However, further studies are needed to demonstrate the necessity of applying a preventive vaccine and to create conditions for reducing the spread of HEV. This review emphasizes the hepatitis E virus and its importance for public health in Europe, the methods of virus transmission and treatment, and summarizes the latest studies on HEV vaccine development.

## 1. Introduction

Worldwide, the hepatitis E virus causes 20 million cases annually, most of which remain asymptomatic. Of those, 3.3 million are symptomatic cases, leading to ~44,000 deaths [1]. This makes hepatitis E the most common cause of viral hepatitis. In European countries, the cases of hepatitis infections caused by HEV are constantly increasing. Between 2005 and 2015, 21,081 symptomatic cases were reported [2]. HEV is causing sporadic and endemic acute viral hepatitis in Europe, with HEV seroprevalence ranging from 0.6% to 52.5% [3,4,5,6]. People who have direct contact with animals (vets, farmers, hunters) have higher seroprevalence rates than the general population [7]. It is known that five HEV genotypes (HEV1-4 and HEV-7) cause hepatitis in humans. Autochthonous hepatitis E in the developed world is caused by HEV-3, which is mainly a porcine zoonosis [8,9]. Sporadic cases of autochthonous HEV have been reported in many European countries after the consumption of raw and undercooked meat products, which is direct evidence of zoonotic HEV transmission [10,11]. In addition, tissue transplantation and blood transfusions have been identified as new pathways for virus transmission [12,13]. In Asia and Africa, the outbreaks are mainly due to contaminated water and lower sanitary and hygienic standards, where HEV-1 and HEV-2 are the prevalent genotypes [14]. HEV-1 and HEV-2 can cause acute hepatitis with liver failure (ALF) or acute-on-chronic liver failure (ACLF). Infections with HEV-3 and 4 are usually asymptomatic, often remain underdiagnosed, and can lead to acute or chronic liver failure in immunocompromised people or patients with solid transplants and pre-existing liver diseases [13,15,16,17,18]. In the general population, the mortality rate ranges from 0.5 to 4%, while in pregnant women infected with HEV, the mortality rate rises up to 30% [19,20,21,22,23]. Although hepatitis E is the leading cause of acute hepatitis, the virus is still not well known to the general public, and sometimes the health authorities do not take adequate measures to limit its spread.

This review provides an overview of the molecular organization of the hepatitis E virus, the virus transmission, and the seroprevalence of HEV in Europe. We also describe the HEV acute and chronic infections and how their clinical manifestation depends on the HEV genotypes. We also review the complications observed in immune-depressed patients and other risk groups and the therapies used in their care. Lastly, we emphasize the need for developing and licensing HEV vaccines.

## 2. HEV Molecular Organization and Taxonomy

The classification of HEV has changed several times since the discovery of the virus in 1978 after the epidemic occurred in Kashmir Valley [24]. The current Virus taxonomy release (July 2021) of the International Committee on the Taxonomy of Viruses (ICTV) classifies HEV in the Family *Hepeviridae* [25]. The given family includes two subfamilies: *Orthohepevirinae* and *Parahepevirinae*. The first subfamily contains four distinct genera: *Avihepevirus*, *Chirohepevirus*, *Paslahepevirus,* and *Rocahepevirus,* while the *Parahepevirinae* subfamily consists of only one genus—*Piscihepevirus* (Figure 1). 

The *Paslahepevirus* genus contains two species, *P. balayani* (formerly known as *Orthohepevirus* A) and *P. alci*. The *P. balayani* species is divided into eight genotypes (HEV A1-8) based on the infected host species. HEV A1 and A2 genotypes are exclusive to humans and are predominantly found in Africa and Asia [26]. Genotypes A3 and A4 are found in humans and some animal species: domestic pigs, wild boar, deer, and rabbits. The expanded host range indicates the high variability of these HEV strains and their zoonotic potential. Both genotypes are responsible for most HEV outbreaks in Europe and East Asia. HEV-A5 and A6 were observed among wild boars in Japan, HEV-A7 was identified in Dromedary camel in the Middle East region, while genotype A8 was identified among Bactrian camel in China [27,28,29]. The *P. alci* species was observed in moose. The *Avihepevirus* genus (formerly known as *Orthohepevirus* B) contains two species: *A. magniiecur* and *A. egretti*. The former was observed among chickens and sparrows, while the latter was detected in little egrets. The *Rocahepevirus* genus (formerly known as *Orthohepevirus* C) contains the *R. ratti* and *R. eothenomi* species, which infect rats and ferrets, respectively. Despite the host preferences, in addition to genotypes A3 and A4, HEV-A5, A7, and A8 from the *Balayani* species as well as the rat-affecting species (C1, C2, C3) of the *Rocahepevirus* genus have zoonotic capabilities [30,31]. The *Chirohepevirus* genus, formerly known as *Orthohepevirus* D) includes three species, *C. eptesici, C. rhinolophi*, and *C. desmoid*, which affect different bat species. The *Parahepevirinae* subfamily contains a single genus *Piscihepevirus* with only one member—*P. heenan*. The indicated species is reported in trout exclusively. The virus shares a low sequence identity with the avian and mammalian HEVs from the *Orthohepevirinae* subfamily, which was the reason for separating it into individual subfamilies [26,32].

### 2.1. Virion Structure of HEV

Infected hosts produce two types of HEV virions. The circulating blood of infected hosts contains the so-called quasi-enveloped (eHEV) particles wrapped in a lipid envelope [33]. Alternatively, the particles secreted with the patient’s feces are non-enveloped (neHEV) [33]. The neHEV variant is known to be more infectious, while the enveloped ones are resilient to antibody neutralization against the viral capsid protein. The experimental expression of capsid proteins in insect cell lines revealed two types of virus-like particles (VLPs). The small version (T = 1) has a diameter of 270 Å, while the native virion (T = 3) is about 25% larger—320–340 Å [34]. Interestingly, the two types are formed by different sets of amino acid residues. The formation of T = 1 VLPs requires amino acids from 126 to 601, while the production of T = 3 VLPs involves residues from 14 to 608, as well as the signal sequence and the N-terminal arginine-rich region. Both VLP types are icosahedral in shape but vary in the number of capsid protein copies contained. T = 1 VLPs contain 60 subunits, while T = 3 comprises three times more, or 180 subunits [35]. The deletion of the N-terminal basic domain of the capsid protein results in the formation of empty T = 1 VLPs that lack the viral genome [36]. 

### 2.2. Genome Organization of HEV 

As stated above, most of the information about the HEV genome is derived from sequences of the *Balayani* species, whereas the exact genome composition of other species needs further research. Current knowledge states that the HEV genome is based on a positive-sense, single-stranded RNA (+ssRNA) with a total length of approximately 7.2 kb. The genome organization of HEV is divided into three open reading frames (ORFs), except HEV-1 has an additional ORF4 (Figure 2).

The RNA features three or four partially overlapping ORFs flanked by 5′ and 3′ untranslated regions (UTRs) [37]. ORF1 is the largest with a total length of 5082 nt and results in a 1692 amino acid polyprotein, crucial for HEV replication. It starts 25 nt after a non-coding region at the 5′ end of the genome and is translated directly from the HEV genome [38]. ORF2 encodes the capsid proteins and has a total length of 1983 nt. It starts 37 nt downstream of ORF1 and ends 65 nt upstream of the Poly A tail [39]. ORF3 demonstrates a slight overlap with ORF2 and translates into a phosphoprotein. The expression begins at the third AUG of ORF3, or nucleotide 5131 [40]. ORF4 has been detected exclusively in genotype A1, where its translation stimulates viral replication. It begins at nt 2835 and ends at nt 3308. The translation of ORF4 is driven by an IRES-like sequence positioned between nt 2701–2787 of the viral genome [41]. The initiation of viral replication and infectivity are heavily dependent on the 5’ UTR 7- methylguanosine cap (7 mG) [42]. The 3’ UTR consists of a PolyA tail, where its U-rich region is used as a potent pathogen-associated motif pattern (PAMP) for retinoic acid-inducible gene I (RIG-I) [43]. Due to the partial overlapping, ORF2 and ORF3 products are expressed from a bicistronic sgRNA (subgenomic RNA). The region between the end of ORF1 and the beginning of ORF3 forms a stem-loop essential for the synthesis of sgRNA.

### 2.3. HEV Proteins and Their Function

#### 2.3.1. ORF1-Encoded Polyprotein

The encoded polyprotein has non-structural properties and is comprised of several potentially functional domains oriented in the following order: Met (methyltransferase), Y domain, PCP (papain-like cysteine protease), HVR (hypervariable region), X domain, Hel (helicase) and RdRp (RNA-dependent RNA polymerase). The Met domain is located at the 5′ end of ORF1 and is known to have guanine-7-Met and guanylyl transferase activities [44]. Although listed as a separate domain, recent studies consider the Y domain an extension of the Met domain. Evidently, this domain influences viral infectivity and replication [45]. The PCP domain acts as an interferon antagonist. It has deubiquitinase activity for RIG-I and TANK binding kinase 1, which plays a crucial role in the evasion of the immune system. The HVR domain is known to enhance viral protein production and replication [46]. The X domain works as an interferon antagonist with the potential to completely block the interferon beta (IFN-b) production and modulate the phosphorylation of interferon regulatory factor 3 (IRF-3) [47]. As expected for a positive-stranded RNA virus, HEV has RNA helicase activity (Hel domain) to ensure the unwinding of the RNA duplex during genome replication. In addition, the Hel domain is engaged in the catalyzing of RNA capping via its 5′ triphosphatase activity. RdRp begins the replication of the complementary RNA strand by binding to the 3′ UTR of the HEV genome. The domain includes a GDD motif which is crucial for enzyme processivity [48]. It is not clear if the ORF1 polyprotein, also known as HEV replicase, needs to be cleaved accordingly into individual units described above or functions as a single protein with replicase activities.

#### 2.3.2. ORF2-Encoded Capsid Protein

This is the basic virion component against which neutralizing antibodies are produced upon infection; however, recently, it was demonstrated that it may have multiple other functions than just forming the virus capsid. Montpellier et al. reported that there are three different forms of the ORF2 capsid protein, ORF2i (infectious), ORF2g (glycosylated), and ORF2c (cleaved) [49]. The ORF2i is the building component of infectious particles; the protein is not glycosylated and is derived from the assembly of the intracellular ORF2 form produced in the cytosol. ORF2g and ORF2c proteins are sialylated and glycosylated. They are not associated with the formation of infectious virions and are found in abundance in the sera of infected patients [50]. Importantly, ORF2g and ORF2c proteins are associated with HEV immune evasion by reducing antibody-mediated neutralization. It is assumed that different pathways are involved in the production of the different ORF2 forms, by differential addressing of the ORF2 protein into the secretory pathway [49] or by a differential translation process [51].

#### 2.3.3. ORF3-Encoded Multifunctional Protein

ORF3 encodes a multifunctional protein with a total size of 113–115 aa residues and is phosphorylated at position Ser71 by ERK (extracellularly regulated kinase). With its function as an ion channel, it plays a critical role in the release of virus particles from the infected cells sharing key structures with class I viroporin. [52]. It also controls the host’s innate immune response, interferes with host signaling pathways, and assists in the “quasi-envelopment” of HEV [53]. These recently discovered characteristics of the protein make it a desirable potential target for the development of antiviral drugs and may improve future vaccines against zoonotic HEV infection. 

#### 2.3.4. ORF4-Encoded Protein

The ORF4 protein in genotype 1 has a molecular weight of 20 kDa and is known for its ability to cooperate with host elongation factor 1 isoform-1 and the HEV proteins RdRp, X, and Hel [41]. The interaction leads to the formation of complex protein, which stimulates HEV RdRp processivity. The role of the ORF4-encoded protein is still under intensive research, and it is yet to be clarified why this ORF is found only in the HEV-1 and if the protein has potential equivalents among the other genotypes [54].

## 3. HEV Transmission and Animal Reservoirs

Typically, HEV is considered a zoonosis in the developed world. Zoonotic transmission occurs via direct contact with infected animals, via consumption of undercooked infectious meat products (mostly pork), and environmental contamination by animal feces since the use of pig manure in agriculture may contaminate water sources and agricultural products [55]. Human-to-human transmission is less common but still possible [56]. Figure 3 displays different HEV transmission models and the risk groups.

### 3.1. HEV as a Zoonosis

Swine-to-human route is a well-known pattern of HEV transmission in developed countries but genotypes 3–8 within *Orthohepevirus* A can also infect other species, such as wild boar, deer, camels, rabbits, and dolphins, enabling additional ways of transmission [57]. HEV-3, the most prevalent genotype in Europe, is believed to spread mainly through the consumption of undercooked pork products from infected pigs and wild boars [58]. However, people with occupational exposure, such as veterinary surgeons, farm workers, and hunters, are shown to be at higher risk of HEV infection than the general population [59,60]. Various local culinary traditions can add additional risk factors for spreading HEV. In this regard, autochthonous acute hepatitis E cases caused by ingesting raw figatelli (a dried, cold-smoked sausage containing ≈30% pig liver) have been reported in France [61,62]. Additionally, several European countries reported HEV in deer species [63,64,65]. Although, recent findings contradict the idea that deer serve as a real viral reservoir since deer can only become infected through close contact with diseased wild boars and their excrement [66]. It is also noteworthy to point out that companion animals, such as dogs, cats, rabbits, and horses, may act as unintentional hosts for HEV and as a source of HEV transmission to humans [67]. 

In many continental settings, rodent populations are very high and diverse and coexist in close proximity with people, pets, and wild and domesticated animals. Growing evidence suggests that rats may be involved in the epidemiology and transmission of HEV. Wistar rats injected with a wild boar-derived HEV-3 strain showed detectable HEV RNA and anti-HEV antibodies [68]. Additionally, HEV-3 has been successfully introduced into a number of rodent cell lines [69]. A rabbit HEV-3ra sequence was found in a Norway rat from Belgium [70], and swine HEV-3 was identified in Black rats in Italy [71]. However, it is still unclear whether rats are actual natural reservoirs of human HEV or merely serve as intermediary hosts.

The fact that HEV from a particular clade of genotype 3 (3ra or rabbit HEV) infect rabbits suggests that this virus is host-specific [72]. Both domestic and wild rabbits around the world carry this particular viral type [73]. The first human infections with rabbit HEV were reported in 2017 in France, where one immunocompetent patient with cirrhosis and four immunocompromised individuals were infected [74]. Also, three Swiss solid organ transplant recipients have been diagnosed with rabbit HEV infections [75].

The main hosts for *Orthohepevirus* C (rat HEV) are mustelids and rodents, and this species was observed in rats at many locations across Europe [70]. In 2022, the first case of *Orthohepevirus* C in acute HEV-infected patients in Europe was reported in Spain. Three of 267 patients with acute HEV infection were positive for rat HEV (HEV-C1). Of the three, one died [76]. Rat hepatitis E in humans has also been reported in France [77]. The outcome of these studies suggests that rat HEV could be considered an emerging disease in Europe. Cases of rat HEV in humans outside of Europe, both in immunocompromised and immunocompetent patients, have been previously reported [78,79]. It has been shown that non-human primates are vulnerable to the rat HEV, suggesting that it could also be transmitted to humans [80]. Rats are a possible natural reservoir for HEV because they interact with people and domestic animals closely and regularly [81]. Therefore, it is important to continue monitoring the prevalence of not only HEV-C1 but also human HEV among the rat population. 

The HEV species and genotypes not covered in detail above are not typically within the territory of Europe yet, but some of them, such as HEV-7, have proven zoonotic potential. In 2016, a chronic HEV-7 infection was described in an immunosuppressed patient from the Middle East who regularly consumed camel meat and milk [82].

### 3.2. Waterborne HEV in Developed Countries

Waterborne HEV has been observed in developing countries, but its waterborne potential in developed countries is still under investigation [83,84]. Genotypes 1 and 2 are responsible for HEV epidemics in developing countries because of poor sanitation. For developed countries, however, these genotypes are not typical, and when observed, it is most likely due to importation from endemic regions during an outbreak [85,86]. However, Fenaux et al. suggest that there might be factors allowing the spreading of HEV by water, even in the industrialized states. For instance, different HEV genotypes or forms may function in various ways, particularly through the water cycle [87]. Water is not the main route of HEV transmission in industrialized countries, but its significance has yet to be fully understood. There is an assumption that rain could wash away fecally contaminated areas such as pig farm backyards and manured fields into other groundwater and open waterways [88]. Certainly, there is evidence in Europe that HEV could contaminate food products by irrigation of crops such as leafy greens and berries [89,90,91]. HEV has also been found in untreated sewage [92,93,94], but the treated water is most likely negative for the HEV genome [93,94,95]. A study in France showed that drinking bottled water was associated with a lower rate of anti-HEV IgG among blood donors [96]. Lastly, HEV prevalence in bivalve molluscan shellfish is another issue of interest regarding HEV in waters. Shellfish are filter feeders that could acquire HEV from contaminated water [97]. The virus could then be transmitted to humans by consuming contaminated shellfish since they are eaten raw or undercooked [98].

### 3.3. HEV Transmission through Blood and Transplanted Organs

Multiple European studies have shown the presence of anti-HEV antibodies and even HEV RNA in blood donors [6,59,99,100]. A review from 2019 summarizes established by the European Union (EU) HEV screening strategies for donated blood. The presented data show an overall HEV-RNA reactivity rate of 1 in 3109 donations (3.2 million donations screened). The rates vary from 1 in 744 donations (France) to 1 in 8636 donations [101]. HEV infections due to blood transfusions have been documented [102]. In one instance, a cancer patient from Japan developed fatal hepatitis after blood-borne HEV transmission [103]. The acquired data raise questions about the safety of blood products since many countries do not screen for HEV [56]. In general, the risk of infection from meat consumption is much higher than the risk of transmission by blood transfusion, meaning that screening blood donors will not stop most cases of HEV transmission. However, it will significantly lower the risk for immunosuppressed and otherwise afflicted individuals who require blood transfusions. Patients with immunosuppressed conditions frequently require multiple transfusions, which raises the risk of transmission [104]. Additionally, there is a significant distinction between ingesting and contracting the virus intravenously. The mucosal barrier in the gut and the acidic environment of the stomach may offer some protection in the case of oral absorption [56]. On the contrary, an in vitro study showed that quasi-enveloped HEV (eHEV) in the blood attaches less effectively to the cell than the non-enveloped HEV virions in stool [105]. Further studies in this regard are needed, but there are pressing arguments that national politics regarding HEV and blood products are also needed [56]. In the Netherlands and Germany, for instance, blood screening for HEV was introduced as a result of risk estimates [106,107]. Since 2012, eight European countries have implemented HEV screening [101]. Although this measure will not greatly affect the HEV distribution among humans, it will protect the most vulnerable to this disease—the immunosuppressed patients.

Blood transfusions are not a substantial pathway for HEV transmission, but they should not be overlooked since the recipients are usually immunocompromised. It is interesting to point out a case describing a liver transplant to a patient positive for HEV-RNA. Re-cirrhosis of the graft happened quickly due to the emergence of a chronic HEV infection [108]. Organ transplants are not tested for HEV RNA, which may become critical for the recipient. Similarly, stem cell transplants also carry a potential risk for HEV infection. A case was reported of a stem cell donor with acute hepatitis E at the time of leukapheresis while being clinically healthy at the time of evaluation [109]. Xenotransplantation is another special issue of interest, especially regarding pigs that are bred for xenotransplant derivates [110].

## 4. HEV Distribution in Europe

### 4.1. HEV in Humans 

According to a systematic review from 2020, 1 in 8 people, or roughly 939 million people globally, have ever had an HEV infection based on anti-HEV IgG positivity [111]. In Europe, the anti-HEV IgG seropositivity rate was 9.31%, the anti-HEV IgM seroprevalence was 0.79%, and the HEV RNA positivity rate was 0.08% (Table 1) [111].

This serological data estimated Europe as more prone to HEV area in comparison with North America and Oceania and less prone in comparison with Africa and Asia, and the results for South America were controversial. On the contrary, molecular studies from Africa did not present evidence for HEV RNA positivity. However, the tested pool was relatively small [111]. Furthermore, it should be taken into consideration that the epidemiology pathway of HEV-1 and -2 (common in developing countries) differ notably from HEV-3 and -4 (common in developed countries) [112]. HEV-3 is the most prevalent genotype in Europe, with three major clades and subgenotypes (HEV-3abjkchi, HEV-3efg, and HEV-3ra) [11]. The geographical distribution of the HEV genotypes is presented in Figure 4. The presented evidence may lead to the hypothesis that HEV-3 is the most widespread on the European continent. Of course, further studies are needed to draw significant conclusions.

A meta-analysis from 2016 determined a significantly higher rate of anti-HEV IgG in Europe—16.90% [5]. The different survey methodologies, periods, and testing assays of both studies must be taken into consideration.

As mentioned above, 1 in 3109 donations in the EU shows HEV-RNA reactivity. Such incidents led to the introduction of screening policies at some locations, but there are not any unitary measures among EU members yet [101]. The European Center for Disease Prevention and Control (ECDC) released a monitoring report that included epidemiological information on HEV in Europe from a 10-year period (2005 to 2015). In this time frame, three EU nations—Germany, France, and UK—reported 16,810 out of a total of 21,018 HEV cases [113]. However, this may be a consequence of more research on HEV being done at these locations; as of now, all three countries have implemented donor screening policies. Another review on the epidemiology of HEV in South-East Europe specifically claims anti-HEV IgG rates in different population groups to be 1.1–24.5% in Croatia, up to 20.9% in Bulgaria, 5.9–17.1% in Romania, 15% in Serbia, up to 9.7% in Greece and 2–9.7% in Albania [114]. A recent study confirmed a 25.9% presence of anti-HEV IgG when 555 blood donors from Bulgaria were tested [59]. Tendentiously, the seroprevalence increases with age [5,114]. 

### 4.2. HEV in Animals 

Undoubtedly, animals play an important role in HEV, HEV-3 in particular, transmission to humans. In Europe, domestic and wild pigs are considered the main animal reservoirs of HEV-3, and the prevalent subgenotypes in humans and animals in certain areas correlate [115]. Reported seropositivity for anti-HEV IgG in pigs varies between 30% and 100% [116,117,118,119,120,121]. A large-scale review of available data up to 2017 showed seropositivity of 30 to 98%, virus presence of 10–100% at the farm level, 8 to 93% seropositivity, and 1–89% at the individual level in pigs [119]. In 2019, a summary of anti-HEV IgG rates in domestic pigs estimated presence levels of 20–54.5%, 29.2–50%, 38.94–50%, and 31.1–91.7% in Serbia, Bulgaria, Romania, and Croatia, respectively. In wild boars, seroprevalence rates were up to 10.3%, 30.3%, and 31.1% in Romania, Slovenia, and Croatia, respectively [114]. Additional data from Bulgaria shows HEV seroprevalence up to 73.65% in 6-month-old pigs and 12.5–40.8% in wild boar [122,123]. Of course, it should be noted that the different studies were performed with different sampling methods and diagnostic tests, but the provided information, nevertheless, gives us a general picture of the prevalence of HEV in *Suidae* across Europe. 

Evidence of HEV infection was found in other mammals. The prevalence of anti-HEV antibodies in 231 cows from Italy was 36.36% [124], but no HEV RNA was detected when milk and feces samples or milk alone were screened from Belgium and Germany, respectively [125,126]. Swine mixed farming constitutes a risk factor for HEV exposure in cattle [127]. However, clear evidence is lacking on the risk of cow milk consumption.

In Germany, HEV antibodies were found in 2.2% (14 from 624) of European brown hares, but no HEV RNA was detected. In contrast, a seroprevalence of 37.3% (47 from 126) was observed for wild rabbits, and 17.1% (28 from 164) of the samples were HEV RNA positive. Genomic analysis clustered 27/28 samples within the rabbit clade of the HEV-3 genotype (HEV-3ra) and the rest into subtype 3g [128]. Farmed and pet rabbits from Italy also were exposed to HEV infection, with seroprevalence rates of 3.40% in 206 farmed rabbits (collected on 7 farms) and 6.56% in 122 pets [129]. 

The rats are an important HEV reservoir—HEV-C1 is well distributed across European rats, and as mentioned above, there is a possibility for transmission to humans [76]. The presence of rat HEV RNA was detected in 63 of 508 (12.4%) Norway rats and Black rats at the majority of sites in 11 (Germany, Hungary, Denmark, Austria, Switzerland, Czech Republic, Belgium, France, Greece, Italy, and Spain) of 12 countries [70]. In Lithuania, RT-qPCR analysis of rat liver samples revealed the presence of rat HEV in 9 of 109 (8.3%) samples [130]. Although all examined samples were negative for HEV-1-4 [70,130], data in the literature supports the possibility of human HEV in rats [131].

The broad host range for HEV and the proven zoonotic potential of the disease imposes the need for a One Health approach. The HEVnet collaborative network is an example of a good practice that aims to understand the HEV sources and transmission routes by molecular epidemiology, thus supporting HEV prevention and control [132].

## 5. HEV Clinical Manifestations 

Hepatitis E infections are usually asymptomatic and self-limiting, resulting in spontaneous clearance of the virus. The clinical manifestation of hepatitis E is closely associated with the immunity of the host (Figure 5). Therefore, the immunological mechanisms of hepatitis E associated with acute and chronic liver failure must be understood [133,134]. 

Furthermore, the clinical manifestation of the different HEV genotypes varies: unlike HEV-3 and -4 infections, most of which are asymptomatic and rarely lead to acute-on-chronic liver failure in elderly or patients with underlying liver disease, HEV-1 and -2 mainly cause acute illness and can lead to acute liver failure or acute-on-chronic liver failure [135]. The immunocompromised individuals infected with HEV-3 and -4 are at increased risk of developing chronic infection [136]. Furthermore, HEV-associated extrahepatic manifestations involving various organs have been reported [137,138,139,140].

### 5.1. Acute Hepatitis

Acute hepatitis E sequentially passes through three phases—pre-icteric (prodromal), icteric, and convalescent period. The pre-icteric period lasts between 7–10 days. It is characterized by nonspecific symptoms—malaise, anorexia, and flu-like symptoms are rarely observed. Some patients experience altered taste; nausea, vomiting, and diarrhea rarely occur; few patients have itchy skin [141]. The symptoms are better presented in HEV-1 and HEV-2 infections and milder in HEV-3 and 4 [142]. At the end of the pre-icteric period, the patients’ urine becomes darker, and their stools are hypo- or acholic [141,143]. The blood levels of alanine aminotransferase (ALT) and aspartate aminotransferase (AST) begin to rise, and by the end of the pre-icteric period, their values vary between 1000–3000 UI/L.

Icteric period–the beginning of the icteric period is characterized by the yellowing of the sclera and skin and the reduction of subjective complaints. Physical examination reveals jaundice of varying intensity, enlarged liver, and enlarged spleen in 10–15% of the cases [143]. The duration of the icteric period varies depending on concomitant conditions, disease severity, and cholestasis development. 

Convalescent period–during the convalescent period, the patient’s general condition improves; the transaminase values normalize in the following 4–6 weeks [144].

### 5.2. Chronic HEV Infection. Reinfection

Chronic HEV infection is characterized by the persistence of HEV RNA in the patient’s serum and/or stool for ≥6 months after an acute infection, accompanied by elevated ALT and AST [13,145]. Chronic HEV infection is primarily observed in immunocompromised individuals. Among those are solid organ transplant recipients [146,147,148]. It has been found that about 60% of the transplanted patients exposed to HEV develop a chronic infection. That risk is potentiated by Tacrolimus therapy and thrombocytopenia [148,149]. Chronic hepatitis E clinically manifests itself with nonspecific symptoms and less often with jaundice. Chronic HEV infection might cause rapid development of hepatic fibrosis, followed by cirrhosis and subsequent liver failure [149,150]. Only a few patients with chronic infection eliminate HEV spontaneously. The rest are usually prescribed antiviral medications. In some cases, liver transplantation is required [150,151,152]. HEV genotype 3 is the leading cause of chronic hepatitis E infection. However, evidence suggests that HEV-7, HEV-4, and rat HEV might cause chronic hepatitis as well [82,153,154,155]. Chronic HEV-1 or -2 infections have not been described [135].

Most of the reported chronic hepatitis E cases are from European countries and not from the USA and Canada [156].

Reinfection with HEV is characterized by elevation of the anti-HEV IgG in HEV IgG (+) positive persons, detection of HEV RNA, and elevation of the transaminases. Reinfection with HEV might cause the development of fibrosis and chronic HEV infection. Reinfection has been described in both immunosuppressed and immunocompetent individuals [157]. 

### 5.3. Complication of HEV Infection

The risk of developing clinically manifested hepatitis increases with age. The most severe complication is fulminant hepatitis which is observed in less than 1% of the patients from the general population and up to 20–30% in pregnant women in their third trimester [158,159,160], mainly in the HEV-1 endemic regions [161,162]. Individuals with pre-existing liver diseases, alcohol abusers, and immunocompromised individuals (organ transplant recipients, people on immunosuppressive medications, people diagnosed with malignant hemopathy, etc.) are at increased risk of developing chronic HEV infection [158,163,164,165,166].

HEV infection (acute and chronic) can also have extrahepatic manifestations, for instance, neurological, renal, hematological, and pancreatic manifestations. They are more common in HEV-3 or 4 than in HEV-1 or 2 infections [141]. Elderly and people with pre-existing liver diseases are at risk for developing acute liver failure or acute-on-chronic liver failure when infected with HEV-3 [135].

#### 5.3.1. Hepatitis E in Pregnant Women

The only distinctive characteristic of the clinical course of acute hepatitis E in pregnant women is the disease severity. In addition to acute liver failure and the rapid progression of cerebral edema, disseminated intravascular coagulopathy is also observed [167,168]. Most of the cases of severe hepatitis E in pregnant women are reported from Asia, and they are due to HEV-1a. In recent years, it has also been reported that HEV-4 infection is associated with pre-term birth [169]. 

#### 5.3.2. Hepatitis E in Newborn

HEV infection in pregnant women is related to miscarriage, pre-term, and/or stillbirth. Newborn children from mothers with HEV infection may develop icteric or anicteric hepatitis, asymptomatic elevation of AST and ALT, and acute liver failure. Some of the infected newborns die with manifested hypoglycemia and hypothermia. Vertically transmitted HEV does not lead to chronic infection [159,160,170].

#### 5.3.3. Hepatitis E in Individuals with Pre-Existing Liver Disease

In people with pre-existing liver disease, a hepatitis E infection may have a severe progression with an uncertain outcome [158,171]. This risk group also includes alcohol abusers—they often experience a severe form of the disease with the development of acute-on-chronic liver failure [171,172]. Severe hepatitis E can also develop in patients infected with other hepatotropic viruses. Indian authors emphasize high percentages of decompensation of concomitant liver diseases during acute hepatitis E infection, with lethality rates reaching 70% [172,173,174,175]. These data do not correlate with the data from industrialized countries, where the fatality rate among patients with underlying liver disease does not appear to increase when infected with HEV [173,174,175]. 

#### 5.3.4. Hepatitis E in Immunocompromised Individuals

Organ transplant recipients, those with malignant hemopathies, and immunosuppressed individuals are at-risk groups for developing chronic hepatitis E. HEV infections in these patients have no typical manifestations. Because these individuals are immunosuppressed, they may have a suboptimal immune response to an HEV infection and not produce antibodies. In such cases, the infection could be confirmed by detecting HEV RNA [176,177,178,179]. Although HIV-positive individuals might develop acute hepatitis E, chronic infections are more common. Clinical manifestations are rarely observed—mainly in patients with CD4+ lymphocytes <200/mL [179]. 

A recent publication reported that seroprevalence among people on hemodialysis is higher than the general population [180]. It is of great importance to evaluate the risk of hospital-acquired infection in these medical institutions [181,182]. 

## 6. HEV Diagnosis Establishment

Diagnosis establishment of the HEV infection is complex and based on clinical, epidemiological, and laboratory data. The etiological diagnosis is confirmed by serological and virological tests. Combining serological and nucleic acid amplification tests is a recommended strategy in the definitive diagnosis of hepatitis E. In immunocompetent individuals, the presence of HEV-specific antibodies in significantly high values is usually sufficient for establishing a diagnosis, whereas HEV RNA detections in serum and stool are the classic tools for determining a diagnosis.

The direct tests include the detection of HEV RNA in serum and stool. These tests have high specificity. However, they are not routinely used. The indirect tests are based on the detection of specific antibodies (anti-HEV IgM, anti-HEV IgG, anti-HEV IgA) in the serum of an infected person [183,184]. A solid-phase assay (sandwich or indirect ELISA) for detecting anti-HEV IgM, anti-HEV IgA, and IgG is used in the diagnosis of HEV [185,186,187]. There is no gold standard for the detection of anti-HEV-specific antibodies, and the different assays often gave diverging results [188]. 

Anti-HEV IgM are markers of a current or recent infection. Anti-HEV IgM appears in the serum during the incubation period and might persist for 3 to 12 months and even longer in some patients [189]. Anti-HEV IgG are markers of resolved or past infection. Anti-HEV IgG appears some days to weeks after anti-HEV IgM and might persist for years. Anti-HEV IgA might be useful markers in individuals with acute hepatitis E that are negative for anti-HEV IgM (10–15%). Simultaneous detection of anti-HEV IgM and anti-HEV IgA supports the diagnosis of acute hepatitis E [190].

Chronic hepatitis E is diagnosed by detecting HEV RNA in serum and/or stool using conventional or real-time RT-PCR. Immunosuppressed patients can present prolonged virus infection that may take up to 6 months, while some immunocompetent ones may develop a protracted form of the infection with a delayed virus clearance of up to 2 years. Diagnosing acute and chronic HEV infection sometimes requires various tests and precise evaluation of the results. The hepatitis E diagnostic tests have continuously improved over the years. The tests used today have high sensitivity and specificity [191,192].

## 7. HEV Therapy

Acute hepatitis E is usually a self-limiting disease that resolves without etiological treatment. Severe forms and fulminant hepatitis are rarely observed in immunocompetent individuals. Treatment with antiviral drugs is indicated mainly for chronic infections. Ribavirin is an antiviral medication used at 600 mg/24 h for 3 months. It has a good therapeutic effect in patients with severe acute hepatitis E, but reports from the past few years show HEV resistance to ribavirin due to mutations [152,193]. Mycophenolic acid is a selective immunosuppressor that inhibits HEV replication. The combination of ribavirin and mycophenolic acid significantly inhibits viral replication [194]. Pegasys and pegIntron are PEGylated interferon alpha 2a and 2b, respectively, and could be used for treating chronic hepatitis E as a monotherapy or combined with ribavirin [195]. The treatment plan for liver transplant patients takes 12 weeks. The adverse effect of the medication is graft rejection [196,197]. Sofosbuvir shows HEV antiviral activity in vitro but has limited efficacy in vivo [198,199]. Although sofosbuvir limits HEV viremia, it does not lead to viral elimination. Zink salts could block HEV replication in vitro by inhibiting RNA-polymerase activity [200]. Silvestrol is a natural compound that blocks HEV replication in vitro, leading to rapid reduction of HEV RNA in the stools of treated mice [201]. T-cell therapy might be an alternative therapeutic possibility [202,203,204]. The last four medications are still under investigation. 

## 8. Development of HEV Vaccine

### 8.1. Inactivated or Modified Live Virus Vaccine

Cell culture-based viral vaccine production is an efficient and cost-effective method. Currently, the advantages of creating an inactivated or modified live virus vaccine for hepatitis E are impractical due to its inefficient replication in most available cell systems. In recent years, a number of cell lines have been reported to support the replication of certain strains of HEV. The virus can adapt and grow into cell culture lines (PLC/PRF/5, A549, A549/D) [48,69,205,206,207], creating an efficient in vitro cell culture system for producing HEV to be used for vaccine preparation.

### 8.2. Recombinant HEV Vaccines Production 

The first HEV genome sequencing, in 1991, opened the door to recombinant vaccine development [208]. Several different genotypes of HEV can infect humans (HEV 1–4 and 7), but up to date, only one HEV serotype is known [34,209]. This fact makes obtaining a protective vaccine against all HEV genotypes possible. Presently, the main efforts aimed at the development of recombinant HEV vaccine are focused on the ORF2 capsid protein, which contains epitopes for neutralizing antibodies [210]. The neutralization epitopes are located in the protruding (P) domain at the C-terminal region of the capsid protein containing aa residues 456–606 [211]. In addition, ORF3 also revealed immunogenic properties. ORF3 is now described as a “multi-function protein with “endless potential” [53]. The protein product of ORF3 was given “endless potential” because it is believed to be involved in the formation of a “quasi-enveloped” HEV virion [212]. Vaccinating experimental animals with ORF3-based purified peptide resulted in potential partial protection [213,214]. The question of whether ORF3-based vaccination would be effective against other virion categories is yet to be answered. 

The ORF2 capsid protein has been successfully expressed in various expression systems (*E. coli*, yeast, insect cells, plants), and their immunogenicity has been evaluated pre-clinically [215,216,217]. Currently, only three of the candidate vaccines have undergone clinical trials in humans. In collaboration, GlaxoSmithKline (GSK), Rixensart, Belgium, and the National Institute of Allergy and Infectious Diseases (NIAID) Bethesda, MD, USA, analyzed the immunogenicity of the HEV-1 ORF2 Sar-55 strain recombinant protein (56 kDa) produced in insect cells. Sar 56 kDa successfully passed clinical trial phase II [218]. Two thousand soldiers in Nepal received three doses of this vaccine, and 100% of the vaccinated formed protective anti-HEV antibodies after the third dose (ClinicalTrials.gov Identifier: NCT00287469) [218]. The second vaccine candidate, the HEV-4 ORF2 p179 (aa 439–617) protein product, expressed in *E. coli,* was developed by Changchun Institute of Biological Products Co., Ltd. (Changchun, China) and passed the phase I clinical trial (Clinical trial NO. CXSL1000041) [219]. The main characteristic of the p179 is that it can self-assemble into virus-like-particles (VLPs) during the process of expression [220]. The p179 candidate vaccine was also assessed for safety and tolerability [219]. The third vaccine candidate is the first approved recombinant HEV vaccine that originated from the HEV-1 capsid protein. It is also known as p239 (aa 368–606), and it was expressed as a non-fusion protein in *E. coli*. The purified HEV 239 assembles as homodimers resulting in virus-like particles. The vaccine contains 30 µg of the purified antigen and 0.8 mg aluminium hydroxide suspended in 0.5 mL buffered saline (ClinicalTrials.gov Identifier: NCT02189603) [221]. It passed the phase III clinical trial and was licensed for use in China in 2011 by China’s State Food and Drug Administration (SFDA). The vaccine has 100% (95% CI, 72–100) efficacy after the third dose and 96% (95% CI, 66–99) after receiving at least one dose [218]. Recently, a p239-based vaccine started a phase I clinical trial in the USA (NCT03827395). The p239-based vaccine was registered with the trade name Hecolin^®^ in China. The efficacy and safety of Hecolin^®^ (NCT02759991) are being evaluated in a phase IV clinical trial with pregnant women who are at higher risk of acute liver failure, pre-term birth, and fetal death during an HEV infection [222]. 

The *E. coli* and insect cells-baculovirus expression systems have been most successful in producing different forms of the ORF2 capsid protein and determining the conditions for its self-assembly into VLPs. Table 2 summarizes the achievements of HEV ORF2 vaccine development in the *E. coli* expression system and insect cells with pre-clinical and clinical trials.

#### 8.2.1. *E. coli* Expression System

One of the first successful strategies for creating an HEV candidate vaccine was based on the fusion protein trpE-C2 containing HEV-1 Burmese strain ORF2 aa 221–660 aa expressed in *E. coli* [223]. The immunized cynomolgus macaques with the fusion protein trpE-C2 challenged with the Burmese strain did not exhibit HEV antigens in its liver or RNA in its stool. Furthermore, the immunized animals challenged with a Mexican strain of HEV-2 exhibited HEV RNA in the stools and HEV antigen in the stools and liver [223]. For the first time, Meng et al. suggested that the HEV neutralization epitope(s) is conformation-dependent and located at the minimal size fragment of aa 166 (aa 439–617) in ORF2 [210]. ORF2 expressed by *E. coli* (N2E as an unfused pattern) can assemble into oligomers and homodimers. The homodimers show a heightened immune response toward serum infected with HEV relative to the monomeric immune response [225]. Ge et al. suggested that the formation of dimers is associated with better exposure to some conformational epitopes, which also determines the strong immunogenicity of the dimers [225]. Rhesus monkeys immunized with purified NE2 displayed specific anti-HEV antibodies and partially protected against HEV infection [225].

The HEV p239 (aa 368–606) and p179 (aa 439–617) vaccines, which are truncated forms of ORF2 (aa 1–660), have been expressed in *E. coli* [219,226,227]. Both vaccines can assemble into VLPs and mimic the native virions [220,233,235,236,237]. The effectiveness of these two candidate vaccines shows that VLP formation is a key factor for strong immune recognition and response and a basic requirement for creating an effective ani-HEV vaccine. Wu et al. demonstrate that E2, which is expressed as a soluble homodimer, is 200 times less immunogenic than the assembled VLPs from p239 [238]. 

#### 8.2.2. Insect Cell-Baculovirus Expression System

Full-length ORF2 HEV-1 Pakistan Sar-55 strain expressed in Sf-9 insect cells shows formed capsid proteins with different sizes (72 kDa, 63 kDa, 56 kDa, and 53 kDa) [239,240]. The study showed that the immunoreactive ORF2 proteins with different sizes are products of a series of protolithic truncations at the N- and/or C-termini [231]. The smallest, 53 kDa product of the ORF2 Sar-55 strain (aa 112–578) was able to form VLPs. By contrast, the 56 kDa ORF2 product (aa 112–607) did not form VLPs [230]. Since the neutralization epitope was mapped between aa 578 and 607, the 53 kDa protein (aa 112–578) did not contain it, while the 56 kDa protein did [241]. Immunizations with the 53 kDa VLPs Sar-55 did not provide adequate protection for rhesus macaques after high-dose challenge [230], while immunization with two doses of 56 kD Sar-55-ORF2 with adjuvant, protected vaccinated monkeys, even after being challenged with heterogeneous genotype 2 or 3 [232]. Further, the recombinant 56 kDa Sar-55 ORF2 was evaluated in human clinical trials (phase I and II), confirming that the vaccine effectively prevents hepatitis E. However, the results of the clinical studies initiated some doubts and questions [242,243,244]. Further studies provide useful information about recombinant HEV VLPs formation. N-terminally truncated HEV capsid proteins (aa 112–660, genotype 1 HEV Burmese strain) expressed in Tn5 and Sf-9 insect cells generated two proteins, 58 kDa (primary transcript) and 50 kDa (further processed protein). Only the 50 kDa was shown to self-assemble into VLPs [233]. Also, ORF2 p495 (aa 112–606) was suggested as suitable for HEV vaccine development, being comparable to p239 (aa 368–606) [229]. Expression of p495 in *E. coli* and Tn5 insect cells displayed comparable, in size and morphology, production of VLPs. Both displayed significant anti-HEV capability, with binding profiles matching that of p239. In addition, p495 elicited immune responses in mice similar to that of p239 [229]. 

#### 8.2.3. Yeast Expression System

The yeast expression system has the advantage of high yield, fast growth, ease of manipulation, low product cost, and eukaryotic posttranslational modification of the recombinant protein production. The yeast *Pichia pastoris* successfully expressed the 112–608 aa sequence of ORF genotype 1 HEV [245]. Purified recombinant protein through Ni-NTA chromatography and density gradient centrifugation was immunologically tested via injection of Balb/c mice, after which was observed significant immunological response because of serum display of high specific IgG titer as well as augmented proliferation of splenocytes in correlation with dosage [245]. Prokaryotic organisms cannot provide proper post-translational modification to ensure appropriate folding and bioactivity of the heterologous recombinant proteins. The eukaryotic organism, such as yeast, is able to perform post-translational modifications (N- and O-glycosylation, disulfide bond formation, removal of signal peptide and etc.). *Pichia pastoris* has been used for studies of the glycosylation pathway in eukaryotes expression systems and the role of N- and O-glycosylation in the immunological properties of heterologous recombinant proteins [246,247]. As additional evidence of VLPs formation in yeast expression systems, yeast functioned as the expression system for HEV-3 and HEV rat protein, where HEV rat protein became altered through glycosylation and displayed self-assembly into VLPs. Self-assembly into VLPs was also exhibited by HEV-3 [248]. Also, methylotrophic yeast *Hansenula polymorpha* was used to express the HEV-4 ORF2 gene encoding aa 112–607 of the capsid protein. ORF2 aa 112–607 product has a molecular weight of 56kD and accumulated up to 12% of total cellular protein [249]. Further, Su et al. reported expression level of HEV VLPs up to 26% of total cellular protein, and the HEV VLPs vaccine had good antigenicity and immunogenicity [250].

#### 8.2.4. Plant Expression System

Plant expression systems are successfully established as an alternative to conventional expression systems, especially in the production of VLP-based vaccines [251,252]. Plants are safe and easily scalable expression systems that can synthesize functional proteins with eukaryotic posttranslational modification [253,254,255,256]. The early studies of HEV vaccine production in plants are based on stable nuclear or plastid genome transformation. *Lycopersium esculentum* was utilized in oral HEV vaccine development [257]. The HEV sequence of interest, pE2 (ORF2 394–604 aa), was cloned and successfully integrated into the tomato plant genome using *Agrobacterium tumefaciens*. The recombinant pE2 protein was expressed at levels of 61.22 ng/g fresh weight (FW) in resultant tomato fruits and 6.37 ng/g to 47.9 ng/g FW in transgenic leaves. Disappointingly, transgenic tomatoes do not accumulate truncated ORF2 at a high level for efficient oral vaccine production [257]. The expression of truncated ORF2 was improved by achieving transplastomic *Nicotiana tabacum cv. SR1* plants. Tobacco plants accumulated pE2 at higher levels (13.27 µg/g FW) compared to transgenic tomatoes. The plant-derived recombinant pE2 peptide induced specific anti-HEV Ab in mice [258]. Orally immunized mice with transgenic potatoes producing truncated ORF2 capsid protein did not elicit specific anti-HEV capsid protein antibody responses [259]. To increase the expression of the HEV ORF2 capsid protein in plants, transient expression vectors were used, which yielded up to 200 µg/g of FW [215,217,260]. Transient expression of the full-length HEV-3 ORF2 (aa 1–660) gene and its N-terminal and C-terminal truncated forms: (aa 1–610), (aa 33–660), (aa 33–610), (aa 110–660), (aa 110–610) constructs produced several immunoreactive proteins with size 97 kDa, 64 kDa, 58 kDa, 53 kDa, and 51 kDa. It should be noted that only expression of the HEV-3 ORF2 (aa 110–610) gene construct leads to the synthesis of a protein that can self-assemble into VLPs with various sizes (19 nm to 31 nm) [215]. Zahmanova et al. demonstrated that N-terminal truncation to aa residue 110- and C-terminal truncation to aa residue 610 is essential for HEV VLPs formation in plants. The plant-derived recombinant protein HEV-3 ORF2 (aa 110–610) VLPs induced high titers of specific IgG antibodies in BALB/c mice after the third immunization with 50 µg VLPs plus Derinat adjuvant [217].

### 8.3. Chimeric Vaccines

Chimeric vaccines are composed of the immunogenic protein from two or more viruses, and they induce an immunogenic response against more than one disease. In recent years, HEV has been a part of dual-vaccination efforts against foot-and-mouth disease (FMDV) [261], hepatitis B virus (HBV) [262], porcine circovirus 2 (PCV2) [263], influenza virus [215,217,264], hepatitis A virus (HAV) [265,266], norovirus (NoV) and astrovirus (AstV) [267], and severe acute respiratory syndrome coronavirus 2 (SARS-CoV-2) [268]. The formation of chimeric proteins can vary based on the chosen host cell expression system, which is also the case in the formation of non-chimeric proteins. *E. coli* is commonly involved in these protocols. *E. coli* was used in the formation of the first HEV-FMDV chimeric protein, where this chimeric protein exhibited the characteristics of water solubility and VLPs self-assembly [261]. The protein was included in a heat-stable vaccine candidate [261]. *Nicotiana benthamiana* successfully expressed HBc-HEV chimeric protein, with spontaneous self-assembly into “ragged” VLPs. Initially, aa 551–607 of HEV-3 ORF2 were inserted into HBcAg, which was previously cloned into the pEAQ-mEL vector [262]. *E. coli* was again employed in the construction of chimeric HEV-PCV2 protein, where the protein was stable even at higher temperatures and exhibited spontaneous formation of spherically shaped VLPs [263]. Of the five chimeric proteins which *E. coli* host cells were able to express, nPCV2cp-p166 was considered the best candidate for vaccine development after considering the following four factors: (1) expression, (2) solubility, (3) folding into VLPs, and (4) post-injection immune response [263]. HEV ORF2 capsid proteins have been used as scaffolds of the M2e influenza peptide or the receptor binding protein (RBP) of SARS-CoV-2. The chimeric genes were transiently expressed in *N. benthamiana* plants using a self-replicating potato virus X (PVX)-based vector. The chimeric protein HEV/M2 was expressed at about 300 μg/g FW, while the fusion HEV/RBD protein was at about 80 μg/g FW. It was found that the chimeric proteins form nanosized VLPs [264].

NoV and AstV are among the leading causes of viral gastroenteritis in humans resulting in significant morbidity and mortality worldwide [267]. Xia et al. reported the development of a trivalent vaccine against HEV, NoV, and AstV by fusing the dimeric capsid P domains of the three viruses. The chimeric vaccine induced significantly higher antibody responses in mice against all viruses compared to the mixed vaccine, which was generated by mixing the three free capsid P domains [269].

Xiang et al. reported the development of a bivalent vaccine against hepatitis A and E infections. Recombinant HAV-HEp148 was constructed as a vector to express a neutralization epitope located at aa 459–606 of the HEV capsid protein. HAV-susceptible cell lines expressed the recombinant virus HAV-HEp148 and the HEp148 protein in a partially dimerized form. Mice immunized with the HAV-HEp148 virus produced strong anti-HAV and anti-HEV-specific antibody responses [265]. 

Gao et al. developed a candidate vaccine by conjugating an immunostimulatory peptide tuftsin to HEV ORF2 (aa 368–607) and partial virus protein 1 (VP1) sequence (aa 1–198) of HAV (HA-VP1). Intranasally immunized BALB/c mice with HE-ORF2-tuftsin + HA-VP1-tuftsin + adjuvant CpG synthesized serum-specific IgG and IgA antibodies against HEV and HAV at the intestinal, vaginal, and pulmonary interface. This candidate vaccine demonstrated the importance of further developing a combined mucosal vaccine against HEV and HAV infections [266].

Chimeric proteins are at the forefront of research, offering a brighter future for combatting HEV and other debilitating conditions via recombinant vaccine technology.

### 8.4. Vectored DNA Vaccines 

Recombinant viral vectors have been used to deliver the complete or truncated HEV ORF2 capsid protein gene and develop a candidate DNA-based vaccine. Trabelsi et al. used adeno-associated virus (AAV) as a vector for the expression of the truncated HEV ORF2 gene coding the capsid protein (aa 112–660). Baculovirus and insect cells were used for the production of rAAV, and the HEV ORF2 (aa 112–660) expression was confirmed [270]. Furthermore, the DNA ORF2 sequence-based vaccines have been evaluated in cynomolgus macaques (cynos) [271]. The full-length of the HEV-1 ORF2 Burmese strain was cloned, resulting in pcHEVORF2. The pcHEVORF2 was used for the genetic immunization of cynos which were subsequently challenged with a heterologous HEV strain (Mexico). Full protection was observed in two of the four DNA-vaccinated animals—the other two developed HEV infection and disease [272]. Further, Deshmukh et al. have used DNA vaccine alone and DNA-prime-protein-boost (DPPB) approaches in mice employing either complete ORF2 or the smaller region of ORF2 containing the neutralizing epitope (NE) [273]. Rhesus monkeys were immunized using the same approach: DPPB-protein encapsulated in liposomes. The DNA vaccine (20 μg) was encapsulated in liposomes and mixed with peptide HEV-1 ORF2 NE (aa 458–607) [274]. After the immunization, macaques were fully protected from challenges with homologous HEV RNA [274]. Arankalle et al. concluded that encapsulating DNA coding for NE (aa 458–607) in liposomes and a protein boost offered the best protection and needed further evaluation as a vaccine formulation [274].

## 9. Conclusions and Recommendations

HEV is the most frequent cause of enterically-transmitted acute viral hepatitis worldwide. Estimates by the World Health Organization (WHO) mentioned in this review indicate that around 20 million people are infected annually, and the overall mortality rate ranges from 0.2% to 4%. In Europe, HEV has been recognized as an emerging problem based on epidemiological data suggesting a steady trend of expanded or even accelerating burden of infection, which cannot be explained only by improved diagnostics and surveillance. The most common risk factor remains the consumption of contaminated and not properly cooked pork or game meat, although exposure to vegetables and shellfish possibly contaminated with sewage has been perceived. Currently, the dynamics of HEV transmission from animal reservoirs are too complex and challenging to control in order to reduce its impact on the food production chain hence the development of successful vaccines is the most practical avenue for the prevention of the disease. During the last two decades, notable progress has been achieved in the development of various recombinant vaccines based on different segments of the ORF2 capsid protein, which are assembled as VLPs mirroring the native surface of the HEV virus particles. The expression and self-assembly of these truncated ORF2 constructs have been described in detail above; a brief summary highlights the use of both prokaryotic (*E. coli*) and various eukaryotic expression systems (insect cells, yeast, plants) demonstrating high immunogenicity of the recombinant products. DNA vaccines, as well as vectored vaccines incorporating fragments of ORF2 into AAV or vaccinia virus, have been used successfully as expression vectors eliciting specific humoral and cellular immune responses. However, only three vaccine candidates have undergone clinical trials in humans; two of these have been discontinued despite being highly immunogenic (HEV p179 and Sar 56 kDa), which seemed at the time as an ominous testimonial of the lack of interest and investment. The third one, Hecolin, also known as HEV 239 (amino acids 368–606 of ORF2), is expressed in *E. coli* as homodimers resulting in VLPs. The short-term vaccine efficacy after three doses was >99%, while the long-term (after 54 months) was 86.8%. It is manufactured by Xiamen Innovax Biotech, Xiamen, China, and was approved in 2011. So far, however, apart from China, it is licensed only in Pakistan, although approval is being pursued in other Asian countries. Further to that, the vaccine has barely been used outside of China (an effectiveness study in Bangladesh and a vaccination campaign targeting a displaced persons camp in Sudan are both ongoing) and has not been yet prequalified by the WHO, without which endorsement it is unlikely to be recommended for prophylactic use or in emergency settings such as outbreaks.

A potential note of concern about Hecolin, as well as for all other HEV vaccine candidates under development, is the observed antigenic variation of ORF2 protein among the different zoonotic and anthropotropic genotypes, albeit it is considered that all of them belong to a single serotype hence achieving cross-protection against infection by a heterologous HEV genotype is expected. Indeed, several studies showed that immunization with the Hecolin vaccine, which is based on HEV-1, granted protection to rabbits upon HEV-4 challenge. However, the degree of cross-protection may vary from full to partial to possibly inadequate among different zoonotic genotypes; at least one study assessing the cross-protective ability among different mammalian HEV genotypes showed only partial protection against HEV-3 [275]. Occasionally even pigs immunized with HEV-3 were not 100% protected upon homologous challenge [276].

The existence of independent genotype-specific and common neutralizing epitopes among HEV-1 and HEV-4 has been well established, confirming the immunogenicity difference between them. Further to that, in light of the considerable genetic heterogeneity of the zoonotic ORF2, notably between HEV-3 and HEV-4, it is conceivable that a single genotype-based vaccine may not be sufficient to provide full cross-protection among recipients. Furthermore, rat HEV (HEV-C1, member of the *Rocahepevirus* genus) was found as a cause of hepatitis in humans [277]. HEV-C1 shows a high divergence from the usual cause of human hepatitis, *P. balayani* (HEV-A). The extent of coverage of *Rocahepevirus ratti* by *P. balayani* containing vaccines is uncertain. They only appear to be partially immunogenic; therefore, an *R. ratti*-based vaccine based on the p239 fragment has been developed [278].

These reflections should not be ascertained as a matter of contention on the topic of current vaccines but rather as a fresh perspective on the future vaccine development, particularly against zoonotic HEV genotypes in line with the compelling need for an internationally recognized, universal HEV vaccine to meet the challenge of the global circulation of hepatitis E virus.

## Figures and Tables

**Figure 1 viruses-15-01558-f001:**
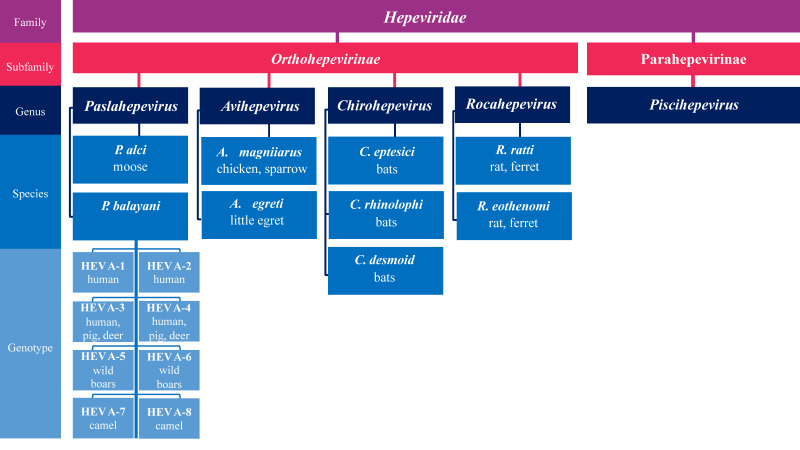
Hepatitis E taxonomy: Hepeviridae family is classified into two subfamilies, *Orthohepevirinae* and *Parahepevirinae*. *Orthohepevirinae* contains four genera: *Paslahepevirus*, *Avihepevirus*, *Chirohepevirus*, and *Rocahepevirus*; Genus *Paslahepevirus* contains two species, *Paslahepevirus balayani* (formerly known as *Orthohepevirus* A) and *Paslahepevirus alci*.; Genus *Avihepevirus* contains species: *Avihepevirus magniiecur* and *Avihepevirus egretti*; Genus *Chirohepevirus* contains species *Chirohepevirus eptesici*, *Chirohepevirus rhinolophi*, and *Chirohepevirus desmoid*; Genus *Rocahepevirus* contains *Rocahepevirus eothenomi* and *Rocahepevirus ratti*. *P. balayani* species is divided into eight genotypes (HEV A1-8).

**Figure 2 viruses-15-01558-f002:**
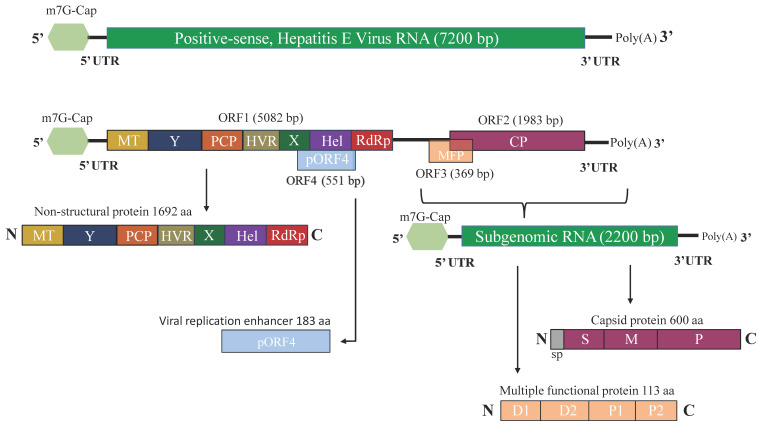
Genome organization of HEV. (a) HEV genomic RNA has three ORFs (ORF1, ORF2, and ORF3), except HEV genotype 1, which has an additional ORF4. ORF1 encodes for non-structural proteins participating in viral replication. ORF2 codes for a capsid protein, ORF3 is translated into a multifunctional protein, and ORF4 is translated into a small protein in genotype 1 with the ability to co-operate with the host elongation factor 1 isoform-1 and HEV proteins RdRp, X, and Hel.

**Figure 3 viruses-15-01558-f003:**
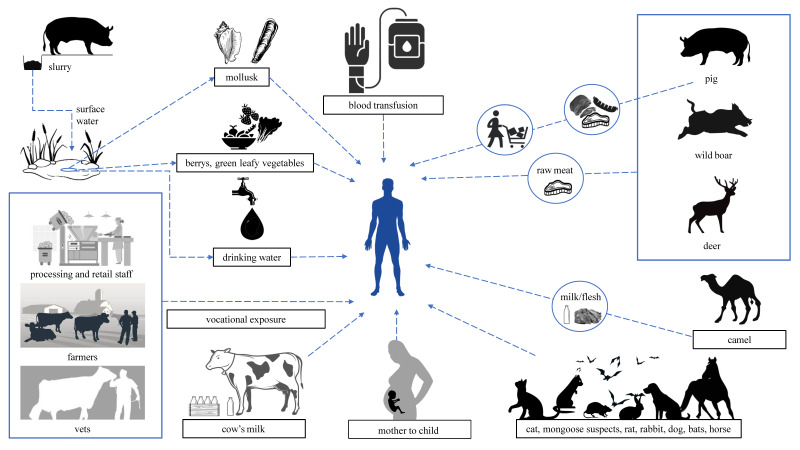
Models of HEV transmission.

**Figure 4 viruses-15-01558-f004:**
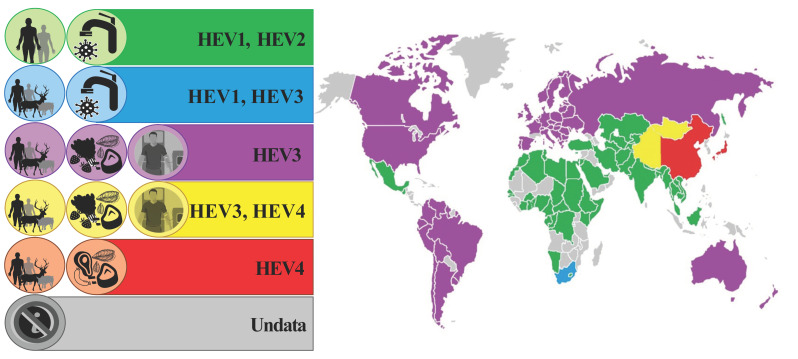
Geographical distribution of HEV genotypes and its reservoirs, and route of transmission.

**Figure 5 viruses-15-01558-f005:**
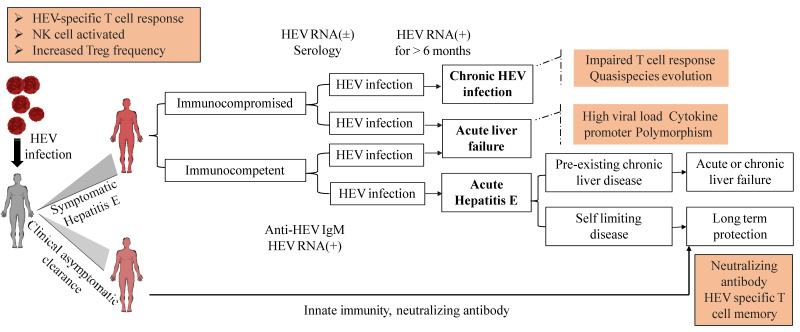
Patterns of hepatitis E virus infection manifestation.

**Table 1 viruses-15-01558-t001:** Hepatitis E seroprevalence and HEV RNA detection in the general population, according to data from Li et al. [111].

		Africa	Asia	Europe	North America	South America	Oceania
**IgG**	% positive	21.76%	15.80%	9.31%	8.05%	7.28%	5.99%
	**n** individuals	22,377	681,373	132,419	71,989	14,586	1563
**IgM**	% positive	3.09%	1.86%	0.79%	0.22%	2.43%	-
	**n** individuals	5001	141,565	146,322	12,197	2680	-
**HEV RNA**	% positive	0.00%	0.93%	0.08%	0.00%	0.18%	0.00%
	**n** individuals	278	727,744	2,441,774	34,761	1054	74,131

**Table 2 viruses-15-01558-t002:** The main achievements of HEV ORF2 expression and vaccine development in prokaryote and insect cells with pre-clinical and clinical trials.

Name	HEV Genotype/ ORF2 Length	Protein Size/VLPs Formation	Expression System	Immunological Studies	References
trpE-C2	HEV-1 Burmese strain 221–660 aa	fusion protein	*E. coli*	Monkeys are fully protected after challenged with a HEV-1 Burmese strain	[223]
E2	HEV-1 394–606 aa	23 kDa homodimer	*E. coli*	Rhesus monkeys are fully protected against infection with a HEV-1 virus.	[224,225]
p239	HEV-1 368–606 aa	30 kDa VLPs	*E. coli*	Passed clinical trial phases II and III, showed 100% efficacy	[226,227]
p179	HEV-4 439–617 aa	20kDa VLPs	*E. coli*	passed phase I clinical trial	[219,228]
p495	HEV-1 112–606 aa	53 kDa VLPs	*E. coli* and Tn5 insect cells	Mouse immunization results showed that p495 VLPs extracted from *E. coli* and insect cells had comparable immunogenicity, as well as p239	[229]
Full-length ORF2	HEV-1 Sar-55 strain (Pakistan) aa 1–660	72 kDa no VLPs 63 kDa no VLPs 56 kDa form VLPs 53 kDa no VLPs	Insect cells	Rhesus macaques immunized with the 53 kDa protein are not fully protected 56 kD Sar-55-ORF2 with adjuvant-induced protection against HEV in monkeys challenged with HEV-1, 2, or 3 strain 56 kDa Sar-55 ORF2 was evaluated in human clinical trials phase I and II.	[230,231,232]
ORF2 fragment	HEV-1 Burmese strain aa 112–608	50 kDa in size forms VLPs	Insect cells	Oral administration of five 10 mg doses of VLPs can confer protection from hepatitis for immunized macaques upon challenge with 10,000 50% monkey infectious dose (MID_50_) of homogenous genotype 1 strain	[36,233,234]

## Data Availability

Not applicable.

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
