# Peer review of "The Re-Emergence of Hepatitis E Virus in Europe and Vaccine Development"

_viruses, 2023, doi:10.3390/v15071558_

Round 1
Reviewer 1 Report
Comments and Suggestions for Authors
Dear Editor,
I have read with great interest the presented review article on hepatitis E virus infection. The manuscript is generally well-written. Please see some minor comments below.
General comments
The abbreviations should be explained when mentioned for the first time and used consistently thereafter.
Line 52: Please correct " ... cause hepatitis in humans."
Figure 1 is of low resolution, please correct it.
Please correct the family and the genera names like in the manuscript text (italic).
Line 183: Please correct " ... neutralizing antibodies are produced upon ... "
Line 260: Please correct " ... to the rat HEV, suggesting ... "
Line 316: Please correct "Blood transfusions are not ... "
Line 334: Please correct "These epidemiological data estimated ... "
Line 407: Please correct " ... manifestation of hepatitis E is ... "
Line 516: Please add some more details on serology tests (ELISA, immunoblot).
Lines 535-536: I suggest using antiviral drugs instead of antiviral medications.
Lines 538-545: ribavirin, mycophenolic acid, and sofosbuvir should be written in lowercase letters.
Lines 558-559: Please correct " ... used for vaccine preparation."
Line 562: Please correct " ... genotypes of HEV can infect ... "
Line 603: Please correct "The E. coli and ..."
Line 609: Please correct E. coli expression system
Line 666: Please correct " ... specific IgG titer ... "
Line 734: Please correct "Nov and Astv are among ... "
Line 745: Please correct "anti-HEV specific antibody response"
Lines 762 and 771: Please correct HEV-1
Line 795: Please correct AAV (the abbreviation has been explained above)
Lines 811-833: Please correct HEV-1, HEV-3, HEV-4, and HEV (the abbreviations have been explained above).
Author Response
Authors’ Response to Reviewer 1
I have read with great interest the presented review article on hepatitis E virus infection. The manuscript is generally well-written. Please see some minor comments below.
Dear Reviewer, we appreciate your valuable comments and your help to improve the quality of the manuscript.
General comments
The abbreviations should be explained when mentioned for the first time and used consistently thereafter.
- Response: Dear Reviewer, thank you very much for your comment about abbreviation and the help to use it properly.
Line 52: Please correct " ... cause hepatitis in humans."
- Response: Done
Figure 1 is of low resolution, please correct it. Please correct the family and the genera names like in the manuscript text (italic).
Response: We made figure 1 with better resolution and made the family and the genera names like italic.
Line 183: Please correct " ... neutralizing antibodies are produced upon ... "
Response: It was corrected.
Line 260: Please correct " ... to the rat HEV, suggesting ... "
Response: It was corrected.
Line 316: Please correct "Blood transfusions are not ... "
Response: Done
Line 334: Please correct "These epidemiological data estimated ... "
Response: It was corrected.
Line 407: Please correct " ... manifestation of hepatitis E is ... "
Response: It was corrected.
Line 516: Please add some more details on serology tests (ELISA, immunoblot).
Response: It was added: “A solid-phase assay (sandwich or indirect ELISA) for detecting anti-HEV IgM, anti-HEV IgA and IgG is used in diagnosis of HEV. There is no gold standard on detection of anti-HEV specific antibodies, and the different assays often gave diverging results.”
Lines 535-536: I suggest using antiviral drugs instead of antiviral medications.
Response: It was change to drugs.
Lines 538-545: ribavirin, mycophenolic acid, and sofosbuvir should be written in lowercase letters.
Response: It was done.
Lines 558-559: Please correct " ... used for vaccine preparation."
Response: “Production was substituted with “preparation”.
Line 562: Please correct " ... genotypes of HEV can infect ... "
Response: It was corrected
Line 603: Please correct "The E. coli and ..."
Response: It was corrected.
Line 609: Please correct E. coli expression system
Response: It was corrected.
Line 666: Please correct " ... specific IgG titer ... "
Response: It was corrected.
Line 734: Please correct "Nov and Astv are among ... "
Response: It was corrected.
Line 745: Please correct "anti-HEV specific antibody response"
Response: Done
Lines 762 and 771: Please correct HEV-1
Response: Done
Line 795: Please correct AAV (the abbreviation has been explained above)
Response: Done
Lines 811-833: Please correct HEV-1, HEV-3, HEV-4, and HEV (the abbreviations have been
explained above).
Response: Done

Reviewer 2 Report
Comments and Suggestions for Authors
This manuscript discusses HEV infection in Europe and vaccine development. The manuscript is written well to review the topic however references cited throughout the text are review papers rather than the actual reports related to the statements and some of the references cited are written in French, German, and Chinese. These references need to replace the reference written in English.
Major comments
Line 35, “ the two countries” needs to describe the specific names of the countries.
In lines 61-62, this statement is misleading. The word “rarely” needs to remove. References used for this statement need to use the findings related to the statement rather than a book chapter.
In lines 63-65, the statement describes the mortality rate related to HEV infection in the general population and pregnant women, however, the reference cited for this statement is a review paper. The experimental observations regarding this statement need to cite rather than the review paper.
In line 100, the authors discussed the finding of genotype 8 in Bactrian camel in China but cited reference is “Structural and Molecular Biology of Hepatitis E Virus”, actual report needs to cite.
For lines 115-117, the reference is missing.
Line 274, the authors described waterborne HEV in the developed counties but a reference for this statement is a review paper rather than a report of the actual finding.
In Figure 5, HEV RNA positivity for >3 months was indicated before the chronic phase, but chronic HEV infection was described as persistent HEV RNA for >6 months in line 438. These descriptions need to be consistent.
Line 455, what is the “chronification”?
Reference 173 is German not English.
Line 519, anti-HEV IgG is resolved or past infection not recent infection.
No references are cited in lines 519-524, 540, 542,581, 583, and 775. Need to provide references for these statements.
Line 527, “elimination” needs to change to “infection”.
In line 557, “ these cells” need to describe specific names of the cell culture system.
The source of reference 196 is not possible to identify.
References 199 and 216 are review papers rather than the original report with findings related to the statements.
A statement in lines 665-665 needs to rewrite. It is confusing to understand.
References 200, 206, 230, and 231 are Chinese papers, not English.
In lines 667-668, the authors need to describe a more specific way rather than “something different”.
Line 669, what are such future studies? Are these future vaccine studies?
Are “Xiang et al” in line 740 and reference 247 the same reference? Or there is a missing reference in line 740?
Author Response
Authors’ Response to Reviewer 2
Dear Reviewer,
Thank you very much for your critical reading and major comment!
Line 35, “ the two countries” needs to describe the specific names of the countries.
Response: Тhere is a recombinant HEV vaccine, but it is approved for use and commercially available only in China and Pakistan.
In lines 61-62, this statement is misleading. The word “rarely” needs to remove. References used for this statement need to use the findings related to the statement rather than a book chapter.
Response: The word “rarely” was removed. The references were updated.
In lines 63-65, the statement describes the mortality rate related to HEV infection in the general population and pregnant women, however, the reference cited for this statement is a review paper. The experimental observations regarding this statement need to cite rather than the review paper.
Response: The references were updated.
In line 100, the authors discussed the finding of genotype 8 in Bactrian camel in China but cited reference is “Structural and Molecular Biology of Hepatitis E Virus”, actual report needs to cite.
Response: The references were updated.
For lines 115-117, the reference is missing.
Response: The reference was added.
Line 274, the authors described waterborne HEV in the developed counties but a reference for this statement is a review paper rather than a report of the actual finding.
Response: The references were updated.
In Figure 5, HEV RNA positivity for >3 months was indicated before the chronic phase, but chronic HEV infection was described as persistent HEV RNA for >6 months in line 438. These descriptions need to be consistent.
Response: The statement was changed: Chronic HEV infection is characterized by the persistence of HEV RNA in the patient's serum and/or stool for ≥ 6 months after an acute infection, accompanied by elevated ALT and AST. Figure 3 was chan
Line 455, what is the “chronification”?
Response: The word was substituted with “chronic HEV infection” for clarity.
Reference 173 is German not English.
Response: The reference was removed.
Line 519, anti-HEV IgG is resolved or past infection not recent infection.
Response: “Recent was substituted with “resolved”.
No references are cited in lines 519-524, 540, 542,581, 583, and 775. Need to provide references for these statements.
Response: The reference were added.
Line 527, “elimination” needs to change to “infection”.
Response: “Elimination” was substituted with “infection”.
In line 557, “ these cells” need to describe specific names of the cell culture system.
Response: The virus can adapt and grow into cell culture lines (PLC/PRF/5, A549, A549/D).
The source of reference 196 is not possible to identify.
Response: The references were updated.
References 199 and 216 are review papers rather than the original report with findings related to the statements.
Response: Reference 216 was substituted.
A statement in lines 665-665 needs to rewrite. It is confusing to understand.
Response: The sentence was rewritten: In addition, p495 elicited immune responses in mice similar to that of p239.
References 200, 206, 230, and 231 are Chinese papers, not English.
Response: MDPI, there is no limit on citing literature in a language other than EN. These articles have abstracts available in English on Pubmed. If the editors consider that References 200, 206, 230, and 231 cannot be cited, we will have to remove the cited information.
In lines 667-668, the authors need to describe a more specific way rather than “something different”.
Response: We rewrote the paragraph; Prokaryotic organisms cannot provide proper post-translational modification to ensure appropriate folding and bioactivity of the heterologous recombinant proteins. The eukaryotic organism, such as yeast is able to perform post-translational modifications (N- and O-glycosylation, disulfide bond formation, removal of signal peptide and ect.). Pichia pastoris has been used for studies of the glycosylation pathway in eukaryotes expression systems and the role of N- and O-glycosylation in the immunological properties of heterologous recombinant proteins.
Line 669, what are such future studies? Are these future vaccine studies?
Response: We rewrote the paragraph: Prokaryotic organisms cannot provide proper post-translational modification to ensure appropriate folding and bioactivity of the heterologous recombinant proteins. The eukaryotic organism, such as yeast is able to perform post-translational modifications (N- and O-glycosylation, disulfide bond formation, removal of signal peptide and ect.). Pichia pastoris has been used for studies of the glycosylation pathway in eukaryotes expression systems and the role of N- and O-glycosylation in the immunological properties of heterologous recombinant proteins.
Are “Xiang et al” in line 740 and reference 247 the same reference? Or there is a missing reference in line 740?
Response: It is the same reference, however, the names of the author were reversed. The reference was corrected.
